# Nutritional, Cardiovascular Health and Lifestyle Status of ‘Health Conscious’ Adult Vegans and Non-Vegans from Slovenia: A Cross-Sectional Self-Reported Survey

**DOI:** 10.3390/ijerph18115968

**Published:** 2021-06-02

**Authors:** Boštjan Jakše, Barbara Jakše, Uroš Godnov, Stanislav Pinter

**Affiliations:** 1Department of Food Science, Biotechnical Faculty, University of Ljubljana, 1000 Ljubljana, Slovenia; 2Barbara Jakše Sole Proprietor, 1230 Domžale, Slovenia; barbara.tursic@gmail.com; 3Department of Computer Science, Faculty of Mathematics, Natural Sciences and Information Technologies, University of Primorska, 6000 Koper, Slovenia; uros.godnov@ikp-dqi.com; 4Basics of Movements in Sport, Faculty of Sport, University of Ljubljana, 1000 Ljubljana, Slovenia; stanislav.pinter@fsp.uni-lj.si

**Keywords:** vegan, non-vegan, diet, nutrition, body composition, cardiovascular health, lifestyle, blood lipids, blood pressure, COVID-19

## Abstract

The study aim was to investigate differences in nutritional, cardiovascular and lifestyle status of ‘health conscious’ subjects. In a partial ‘lock-down’ during the COVID-19 pandemic period, we performed a web-based, cross-sectional study. We compared 80 self-selected subjects (51 vegans, 67% females, and 29 non-vegans, 55% females, *p* = 0.344). Nutritional status was assessed by bio-electrical impedance and standardized food frequency questionnaires (i.e., contribution of nutrients from foods and supplementation, combined and separate). Serum lipid concentrations and blood pressure (BP) were assessed from annual or initial examination reports, while sociodemographic, economic, and lifestyle statuses were obtained by standardized questionnaires. Finally, a multivariate linear regression model was used to estimate the relationship between total fiber and saturated fatty acid (SFA) intake and low-density lipoprotein cholesterol (LDL cholesterol) values. The vegans had a significantly lower body mass index (22.8 ± 2.4 vs. 26.6 ± 3.6 kg/m^2^, *p* < 0.001) and body fat % (19.3 ± 7.3 vs. 25.8 ± 8.2%, *p* < 0.001) than the non-vegans. There were significant differences between vegans and non-vegans in energy intake, and most macronutrient (10/12) and micronutrient (15/23) intakes in units/day. Both diets were well designed, with high fiber and low SFA and free sugar intake but remained insufficient in n-3 long chain polyunsaturated fatty acids (for vegans), vitamin D, calcium, sodium (for vegans) and iodine. Vegans also had a significantly lower lipid profile and BP than non-vegans, except for high-density lipoprotein cholesterol. However, both groups met targeted recommendations. Furthermore, fiber and SFA intake and age explained 47% of the variance in LDL cholesterol. In conclusion, ‘health conscious’ vegans and non-vegans with comparable lifestyle statuses had significant differences in dietary intake, body composition and cardiovascular health status.

## 1. Introduction

Over the last decade, interest in research on the effects of vegan diets (i.e., strict plant-based diets) on various health benefits has increased. Vegan diets are consistently associated with improved health outcomes (i.e., reduced risk for common chronic diseases) and has been recognized as beneficial by various professional associations [1,2,3,4,5,6,7]. However, there is still a debate regarding whether this is due to the diet quality per se or a possible healthier lifestyle status of vegans compared with non-vegans. Prior studies did not systematically evaluate well-designed vegan and non-vegan diets; therefore, the potential differences in body composition and health outcomes are not yet completely understood [8,9]. Generally, it has been well established that healthy and active lifestyles have a pivotal role resulting in many health benefits [10].

The amount of data comparing the dietary intake and health status of vegans and non-vegans is greatly increasing. Multiple cross-sectional studies that compared the dietary intake between vegans and non-vegans have revealed differences in many nutrients; some are beneficial for vegans (i.e., higher intake of fiber and lower intake of saturated and total fat) but some are of concern (i.e., lower intake of protein, n-3 long chain polyunsaturated fatty acids (eicosapentaenoic acid (EPA) and docosahexaenoic acid (DHA)), vitamins B_12_ and D, iodine, iron and zinc) [11,12,13,14,15,16,17].

Furthermore, healthier plant-based diets (e.g., diets higher in plant foods and lower in animal foods or excluding all animal foods) were found to be associated with lower cardiovascular (CV) health risk and all-cause mortality, while less healthy or unhealthy plant-based diets were found to be associated with higher CV health risk and all-cause mortality [8,9]. Importantly, it is well accepted that lifestyle interventions are cornerstones in the prevention and management of CV diseases [18,19]. Therefore, the health benefits observed in vegans compared to non-vegans may also be partly attributed to their overall healthier lifestyles (i.e., non-dietary factors, such as differences in alcohol consumption, smoking, physical activity, and stress management) [17,20,21,22].

There is a lack of a standardized definition of what well-designed vegan and non-vegan diets actually mean in terms of macronutrient composition. The authors of the present paper therefore suggest that well-designed vegan (as used by research that has shown noteworthy results [23,24,25,26]) and non-vegan diets (as proposed in government guidelines on healthy eating by the Ministry of Health of Slovenia [27]) should limit total fat intake to up to 20% and 30% of energy, respectively (i.e., saturated fat intake ≤10% of energy for both), fiber intake should be above 60 and 30 g/d, and both diets should limit free sugar intake to ≤5% of energy. The effects of dietary cholesterol on blood lipids remains a matter of debate [28,29,30]. Notably, due to the recommendation of the Ministry of Health of Slovenia, the consumption of five eggs per week in Slovenia is considered part of a healthy diet [31]. Therefore, for the purpose of our study, it is reasonable to suggest that a non-vegan diet limit overall cholesterol intake (i.e., from eggs and other animal-based sources) <300 mg for healthy adults. Likewise, the current dietary guidelines, as part of a healthy non-vegan diet, recommend eating fish up to twice a week, mainly as substitutes for red and processed meat (i.e., which is reflected in sufficient intake of EPA and DHA) [32]. Moreover, both recommended dietary patterns should consist of whole-food sources (i.e., unrefined and minimally processed) as much as possible. For a more definitive conclusion, further studies are needed to compare individuals with dietary patterns that are well designed who have similar healthy and active lifestyles and then examine their effects on body composition and CV health. This area remains understudied and appropriate for further research.

Nutritional and CV status is normally characterized by dietary intake, anthropometry and/or body composition, selected biochemical tests (e.g., lipids) and blood pressure (BP) assessments. Therefore, the aim of the present study was to provide novel insights into whether vegan diets compared with non-vegan diets among ‘health conscious’ adults and in sociodemographic- and lifestyle-matched groups translate into differences in body composition and CV health status. We hypothesized, since we expected to compare matched groups in terms of lifestyle status, that the differences in body composition and CV health status would be due to well-designed dietary patterns. Furthermore, we expected to see less or no pathology among these non-vegans.

## 2. Materials and Methods

### 2.1. Study Design and Eligibility

The web-based, cross-sectional study was conducted in Slovenia during the COVID-19 pandemic period (i.e., between 22 May and 5 August 2020), when restriction of movement was limited to the area of the municipality (i.e., partial ‘lock-down’). We investigated the nutritional and CV health status of self-selected ‘health conscious’ (e.g., a healthy and physically active lifestyle with a healthier diet) vegans, and non-vegans with a self-report study design as the only ethical means to protect the health of the subjects during the pandemic period. Furthermore, the statistician, as an independent researcher who set up the web-based system, ensured that no one had access to the data before the end of the study.

The study was conducted in accordance with the Declaration of Helsinki. Participants were not remunerated financially, and no external research funds were required to conduct the study. This study protocol was reviewed and approved on 19 May 2020, by the Ethical Committee in the field of sport in Slovenia (No. 7: 2020/312), while the trial was registered at https://clinicaltrials.gov (NCT04379622) (first posted on 7 May 2020).

Each adult individual encountered the study invitation via social media or email announcements by certified health/fitness coaches and community leaders regarding healthy and active lifestyles and, after reading the accompanying text, signed an informed consent form for inclusion in the study. Each participant self-identified their current diet by choosing from one of several options (i.e., omnivore, vegetarian (‘lacto’ (e.g., included dairy products), ‘ovo’ (e.g., included eggs), ‘pesco’ (e.g., included fish and seafood), ‘lacto + ovo’ (included eggs and dairy products) [1]) or vegan). Importantly, we checked their dietary choices with actual food intake (to avoid possible discrepancy of the defined dietary pattern with the actually consumed one), thus avoiding potential errors. The subjects had to complete the set of questionnaires that assessed sociodemographics and motives for chosen dietary patterns, nutritional status, CV health, and lifestyle status. We included self-selected ‘health conscious’ participants from Slovenia who voluntarily and anonymously decided to participate in the study according to the inclusion/exclusion criteria.

### 2.2. Subjects

During the COVID-19 pandemic period, we targeted inviting approximately 10,000 people through health/fitness coaches or sports and recreational groups integrated across the country that were known to provide services related to healthy and active lifestyles and a healthy diet, which also included a vegan diet community. Importantly, in the invitation process, we did not directly include ideologically oriented vegan communities through social media. The invitation to potential subjects included (i) the aim of the study, (ii) the components of the study (e.g., outcomes), (iii) the inclusion and exclusion criteria, (iv) the explanation that the study is anonymous, and (v) the assurance that the data will be used solely for the purpose of this research.

The inclusion criteria for qualifying for the study were as follows: adult (older ≥18 years), informed consent to participate in the study, either sex, healthy and active lifestyle (e.g., regularly performing workouts ≥ three times/w based on regular general physical activity) and healthy diet (e.g., eating ≤ three refined items of food per week) with no current BMI limitations. Furthermore, participants should have had the same (current) dietary pattern for ≥1 year, had body composition status measured within last 90 days in Tanita body composition analyzer (see further explanation below in Section 2.3.2), had lipids and BP measures (e.g., from annual or initial examination report within last year). To truly analyze the relevant sample, we used several exclusion criteria: pregnancy or lactation, competitive or top-level athletes, major musculoskeletal restrictions, active common chronic diseases (i.e., CV diseases, cancer, autoimmune diseases), current use of medication affecting plasma lipids or BP and loss of ≥5 kg within the previous year.

Finally, the online invitation with inclusion/exclusion criteria that was part of the survey was read by 809 people, the research survey was opened by 192 people, and 94 people started to complete the survey. In the final analysis, we were able to enroll 80 adult participants who completed the survey without missing data (100%). The final sample was divided based on dietary patterns into two groups: vegans and non-vegans (omnivores and lacto-, ovo- and pesco-vegetarians; Figure 1).

### 2.3. Outcomes

#### 2.3.1. Sociodemographics, Economic Status and Motive for Following the Dietary Pattern

Regarding sociodemographics and economic status, we adapted the questionnaire provided by the National Institute of Public Health of Slovenia [33]. The included variables were sex, age, partner status, living environment, employment status, education, income, and smoking status.

Furthermore, to evaluate the motives for following vegan and non-vegan dietary patterns, we provided a questionnaire adapted by the investigators. We asked the participants to rank 8 different motives for individual dietary patterns, where 1 was the least important and 8 was the most important: (1) health, (2) body mass (BM) management/appearance, (3) environmental concerns, (4) religious reasons, (5) affordable dieting, (6) convenient dieting, (7) animal ethics and (8) satiety/no hunger. For each motive, we calculated the average scores, ranked them, and compared them between the two subject groups.

#### 2.3.2. Anthropometric and Body Composition Measures

The anthropometric and body composition measures (e.g., via bioelectrical impedance body composition monitor (Tanita Corporation, Tokyo, Japan)) that were asked from participants were body height, body mass (BM), maximal lifetime body mass (not necessarily measured), and body fat percentage (BF%). Of note, in Slovenia, Tanita body composition analyzers are the most commonly used in fitness centers, faculties and hospitals and among health/fitness coaches [34]. The anthropometric and body composition indices included body height, body mass (current and maximal lifetime), BMI (current and maximal lifetime) and BF%. Body mass index (kg/m^2^) was calculated from body height and body mass. Of note, we asked participants to provide results that were measured according to manufacturer’s recommendation (i.e., not to eat or drink at least 1 h, not exercise for at least 24 h, not urinate for at least 30 min and females not measured before and after their menstrual cycle) [35]. Further, we compared obtained results of body composition indices with recommended targets for BMI and BF% [36,37].

#### 2.3.3. Dietary Intake

To assess dietary habits in the past year, we used a manual monitoring technique, a 52-item qualitative food frequency questionnaire (FFQ) that was based on a validated 50-item FFQ [16,38]. The FFQ was translated from the original Dutch language into the Slovenian language by a professional translator and already used in Slovenia. This FFQ included nine different frequency categories ranging from ‘never’ to ‘more than three times a day’. The usual food intakes derived from the FFQ were calculated by multiplying the frequency of consumption of specific foods by a standard portion size for each food (as proposed by National Institute of Public Health of Slovenia [39] and by the amount of nutrients present in 1 gram. The daily dietary intake was calculated by summing the nutrient content of each food item. The FFQ has been tested and validated for assessing food consumption with 7-day estimated diet records [38] and had previously been used in Slovenia in two female populations of athletes [40].

For evaluation of dietary intake, we used a dietary software program, Open Platform for Clinical Nutrition (OPEN) [41], which is a web-based application developed by the Jozef Stefan Institute from Slovenia [42] and supported by the EuroFIR AISBL [43] and the European Federation of the Association of Dietitians (EFAD). The food intake data (from the FFQ) were used in the assessment of energy and nutrient intake and in the assessment of the frequency of food group intake. Importantly, based on the FFQ alone, it was impossible to estimate iodine intake from particular food preparation methods (i.e., added iodized salt), so iodine intake is shown only from food sources per se and supplementation use (separately and combined). Furthermore, using the unique FFQ and OPEN system, we were able to accurately distinguish free sugars from added sugar or total sugar. Finally, regarding food data entry, we used a manual method and double checked it to avoid any possible errors in such a small sample. All supplementation (i.e., dietary supplements, protein shakes/meal replacement (MR), and sport drinks; hereinafter referred to as supplementation) were included in the evaluation of dietary intake.

We calculated data on nutrients (first), compared the nutrient intake from the sources of supplementation (second), and estimated the frequency of food intake, meal timing and frequency of eating main meals (third). Furthermore, data on the average nutrient intake from foods and supplementation were evaluated in combination (i.e., foods plus dietary supplements together) for both sexes together due to the limitation in the number of subjects per group. However, we included information on the gender distribution in each group of participants in the descriptive analysis (i.e., found no significant differences). Finally, we calculated data regarding the dietary intake of participants expressed as kcal/day (energy), in units/day (i.e., in g/day (macronutrients), except for dietary cholesterol (mg/day), water intake (L/day), and micronutrients (mg and µg/day)) and % of daily energy intake (macronutrients).

The intake of nutrients of greater importance and concern were compared with recommendations for adults (i.e., with the Central European (German (D), Austrian (A) and Swiss (CH) D-A-CH) reference values [44], except for free sugar intake with the UK Scientific Advisory Committee on Nutrition (SACN) recommendation (<5% of daily energy intake) [45] and for EPA plus DHA intake with the Dietary Reference Values of the European Food Safety Authority [46]. Of note, D-A-CH references values are valid in Slovenia [27].

#### 2.3.4. Cardiovascular Health Status

Lipids and BP were obtained from annual or initial examination reports conducted within the year prior to this study (≤1 year). Appraised CV health factors were total cholesterol (S-cholesterol), high-density lipoprotein (HDL cholesterol), LDL cholesterol, triglycerides and BP. The lipids (LDL cholesterol and triglycerides) and BP were further compared with the recommended guideline targets for primary CV disease prevention [47,48]. As we expected high variability in fiber and SFA intake and LDL cholesterol levels between the groups, we estimated the relationship between fiber and SFA intake and LDL cholesterol.

#### 2.3.5. Lifestyle Characteristics

To assess physical activity (PA), inactivity and time spent using passive transport during the previous 7 days, we used the self-administered Long International Physical Activity Questionnaire (L-IPAQ) [49]. All study subjects were instructed to maintain their pre-existing PA. We emphasized the following components of the L-IPAQ: (1) time (in minutes) spent travelling in motorized transport (e.g., car, bus, train); (2) time (in hours) spent sitting during the weekdays; (3) time (in hours) spent sitting during weekend days; (4) average daily time performing low-intensity PA (in minutes; equivalent to 3.3 metabolic equivalents (METs) or walking); (5) average daily time of moderate-intensity PA (in minutes; equivalent to 4 METs); (6) average daily time of high or vigorous-intensity PA (in minutes; equivalent to 8 METs); and (7) total L-IPAQ score [50]. Components 4–6 were related to transportation, housework/gardening, recreation, sport and leisure PA.

METs are multiples of the resting metabolic rate to yield a score in MET minutes. The total number of MET minutes of PA per week was calculated as a sum of the MET minutes achieved in each category (walking-equivalent, moderate-intensity PA and vigorous-intensity PA). Vigorous-intensity PA was defined as achieving a minimum total PA of at least 1500 MET minutes/week on at least 3 days per week or 7 days per week of any combination of walking-equivalent, moderate-intensity PA or vigorous-intensity PA to achieve a minimum total PA of at least 3000 MET minutes/week [51]. Moderate-intensity PA is defined as 5 or more days per week of any combination of walking-equivalent, moderate-intensity PA or vigorous-intensity PA to achieve a minimum total PA of at least 500 MET minutes/week [52]. These amounts of vigorous-intensity PA and moderate-intensity PA are equivalent to the PA guidelines stating that at least 75 or 150 min/week of vigorous-intensity PA or moderate-intensity PA, respectively, should be achieved [52,53]. The L-IPAQ had a correlation of 0.8 in an assessment of test–retest repeatability [54].

Regarding sleep quality and patterns, we used 19 self-rated questions from the Pittsburgh sleep quality index (PSQI) questionnaire [55]. Overall, we found nine components of the PSQI to be important for our study, while the scores of seven components as per the scoring rules were calculated as a total score (i.e., global sleep quality), namely, (1) subjective sleep quality, (2) sleep latency, (3) sleep duration, (4) sleep efficiency, (5) sleep disturbance, (6) use of sleep medications and (7) daytime dysfunction. The total score range was from 0 to 21, where a higher score indicated worse sleep quality and >5 indicated poor sleep quality [55]. These lifestyle characteristics questionnaires had previously been used in the adult Slovenian population [56].

### 2.4. Statistical Analysis

Statistical analysis was performed using R 4.0.3 with the dplyr [57], ggplot2 [58] and arsenal [59] packages. Dplyr was used for data transformation, ggplot2 for data visualization and arsenal for statistical calculations. For numerical variables, t-tests were used, and the Mann–Whitney U-test was used where the data were not normally distributed. Normality was checked with Shapiro–Wilk’s test. For categorical variables, we used the chi square test, and when the expected frequency in a cell was less than 5, we used Fisher’s exact test.

Multivariate linear regression was used to estimate the relationship between fiber and SFA, intake and LDL cholesterol, adjusted for age, sex and smoking. However, only age from these adjustments was statistically significant. The adjusted R^2^ model was used to assess the contribution of dietary fiber in explaining LDL cholesterol. The model was checked for linearity and homoscedasticity. The threshold for statistical significance was *p* < 0.05. No missing data were present. No sensitivity analysis was performed. Data are presented as the means (standard deviation).

## 3. Results

### 3.1. Sociodemographics, Economic Status and Motive for Following the Dietary Pattern

The vegan and non-vegan groups included 51 and 29 subjects (34 (67%) vs. 16 (55%) females and 17 (33%) vs. 13 (45%) males, *p* = 0.344). The average ages of the vegans and non-vegans were 46 ± 9 vs. 57 ± 10 years, respectively, *p* < 0.001. There were no other significant differences between the vegans and non-vegans in sociodemographic and other characteristics. Complete descriptive statistics by group are summarized in Appendix A.

Furthermore, the vegans and non-vegans followed their dietary pattern primarily due to health motives (score: 6.7 ± 1.4 vs. 5.5 ± 2.1, *p* < 0.001). The second motive for the vegans to follow a vegan diet was BM management/appearance, whereas for non-vegans, it was satiety/no hunger. Importantly, 74% of the total sample ranked health motives as the most important motive. Eating according to the selected dietary pattern for convenience was ranked significantly higher by the non-vegans than vegans (4.4 ± 1.7 vs. 3.6 ± 1.5, *p* = 0.043). Other motive comparisons were not significantly different between the groups. Motive scores for vegans and non-vegans are presented in Appendix A.

### 3.2. Anthropometric and Body Composition Status

Our results suggest that the vegans had a more favorable body composition status than the non-vegans. All anthropometric and body composition variables were significantly lower in the vegans than non-vegans, with the exception of body height. The mean current BMI and BF% of the vegans and non-vegans were 22.8 ± 2.4 vs. 26.6 ± 3.6 kg/m^2^ (*p* < 0.001) and 19.3 ± 7.3 vs. 25.8 ± 8.2%, *p* < 0.001), respectively. In terms of obesity classification, the current average vegan BMI was within the normal range, whereas the average BMI of the non-vegans was in the overweight range [36]. Interestingly, ‘their worst’ condition in terms of maximal lifetime BMI for the vegans and non-vegans averaged in the overweight and obese ranges (26.6 ± 3.7 vs. 30.3 ± 5.3 kg/m^2^, *p* = 0.001). The anthropometric and body composition status are presented in Table 1.

### 3.3. Dietary Intake Status

#### 3.3.1. Intake of Energy and Macronutrients (From Foods and Supplementation)

Table 2 presents the intake of energy and macronutrients of the vegans and non-vegans. The mean energy intake was significantly higher in the vegans than in the non-vegans (2399 ± 507 vs. 1998 ± 419 kcal, *p* < 0.001). In addition, the mean macronutrient composition of the food intake of the vegans and non-vegans was 20 and 33% fat, 57 and 43% carbohydrate, 6 and 3% fiber, and 16 and 20% protein, respectively.

The macronutrients that were not significantly different between the groups in units/d were free sugar and protein intake. The macronutrients that are often of concern in vegan diets, such as protein, were adequate at 1.5 g/kg BM/d [44], while EPA plus DHA were found to be below the recommended levels in vegans only (184 ± 288 vs. the recommended 250 mg/d) [46]. However, macronutrients that are also often of concern in non-vegan diets if not well designed (i.e., low fiber intake and high SFA (≤10% of daily energy intake) and free sugar (<5% of daily energy intake) intake) were all within the recommended ranges [44,45]. However, the cholesterol intake of the non-vegans was 385 ± 507 mg/d, which is above the recommendation (<300 mg/d) [44]. Importantly, the mean fiber intake of the vegans and non-vegans was 75 ± 16 vs. 34 ± 10 g/day (*p* < 0.001), which was adequate [44]. Finally, the vegans had significantly higher total water intake than the non-vegans (2.6 ± 0.6 vs. 2.2 ± 0.6 L/d, *p* = 0.004).

#### 3.3.2. Intake of Micronutrients (From Foods and Supplementation)

Overall, the vegans outperformed the non-vegans in most micronutrients (15/23) in units/day. Importantly, the intake of micronutrients that are often of concern in vegan diets are vitamins B_12_ and D, calcium, iron, zinc and iodine. The intake of these micronutrients was significantly higher in vegan diets, except for iodine, where the intake was similar. Further, the intake of vitamin B_12_ of the vegans and non-vegans was higher than recommended (23 ± 72 and 5.6 ± 4.2 µg/d, *p* < 0.001, vs. recommended 4 µg/d) [44]. However, vitamin D intake was similar for both analyzed dietary patterns but lower than recommended (7.0 ± 17.6 and 6.8 ± 5.4 µg in the vegans and non-vegans, respectively, *p* < 0.001, vs. recommended 20 µg/d) [44]. Importantly, insufficiencies were found in both diets for calcium (979 ± 281 and 826 ± 324 mg/d, *p* = 0.038, vs. the recommended 1200 mg/d) and iodine (112 ± 60 and 96 ± 58 µg/d, *p* = 0.238, vs. 150 and 180 µg/d for females and males, respectively) [44]. Furthermore, sodium intake of the vegans, without taking into account the intake of (iodized) salt, did not meet the recommendation, whereas sodium intake of the non-vegans exceeded the recommendation (e.g., 1500 mg/d) [44]. The micronutrient composition of the food intake (13 vitamins, 6 minerals, and 4 trace elements) is presented in Table 3.

#### 3.3.3. Supplementation

Based on the FFQ, 98% of the vegans and 87% of the non-vegans consumed supplementation (e.g., from 32 different companies). Seventy-one percent of the vegans and 3% of the non-vegans consumed isolated forms of vitamin B_12_, while 27% of the vegans and 43% of the non-vegans consumed vitamin B_12_ within multivitamin dietary supplements.

Omega-3 dietary supplements (i.e., EPA and DHA) were consumed by 67% of the vegans and 47% of the non-vegans. Importantly, EPA and DHA dietary supplements were found to be only sea sources (i.e., fish or shrimp). During the study period (e.g., spring and summertime), the isolated form of vitamin D_3_ dietary supplement was consumed by 22% of the vegans and 7% of the non-vegans.

Furthermore, protein shakes/MR (e.g., vegans consumed plant-based, whereas non-vegans consumed dairy-based or plant-based protein sources) were consumed by 31% of the vegans and 30% of the non-vegans. Last, other dietary supplements (e.g., herbal teas, algae, vitamin C or magnesium, fiber beverage) or sport drinks were consumed by 80% of the vegans and 63% of the non-vegans.

Importantly, the vegans and non-vegans did not consume significantly different amounts of energy or macro- or micronutrients from supplementation. However, there was a significant difference in the source of protein supplementation. As expected, the non-vegans consumed significantly more animal-based protein (e.g., whey) than the vegans who did not consume any animal-based protein from supplementation. Moreover, the amount of protein supplementation for the vegans and non-vegans was rather low (4.5 ± 6.6 vs. 3.8 ± 5.8 g/d, *p* = 0.776, or 0.06 vs. 0.05 g/kg BM/d, *p* = 0.643, respectively).

Typically, there are three nutrients of concern that are suggested for supplementing conventional diets, especially the vegan diet, namely, EPA/DHA, vitamin B_12_ and vitamin D. However, vitamin B_12_ (5.3 ± 4.2 µg/d compared to the recommended 4 µg/d) and EPA plus DHA intake (355 ± 320 mg/d compared to the recommended 250 mg/d) reached recommended levels by the non-vegans with foods alone [44,46]. Furthermore, vitamin D intake by the non-vegans from food alone was 5 ± 3 µg/d and thus insufficient, whereas for vegans, sun exposure or supplementation is the only option. The proportion of total nutrient intake from supplementation is presented in graphic form in Appendix A(macronutrients) and Appendix A (micronutrients).

#### 3.3.4. Frequency of Food Intake and Meal Timing and Frequency

Overall, the vegans consumed significantly more from 11 out of 16 plant-based food groups. In line with dietary intake, both groups ate mostly whole-food, plant or animal-based diets that were minimally refined and ultra-processed. In brief, various sweets, fried potatoes, white bread and others (i.e., ketchup, margarine, dressings) were occasionally consumed. Furthermore, we also found a significantly higher intake of all alcohol drinks among the non-vegans. However, overall absolute alcohol intake among the non-vegans was rather low and infrequent, whereas the vegans did not report meaningful alcohol intake. The complete frequency of food intake comparison between vegans and non-vegans by food group is presented in Appendix A.

The average breakfast, lunch, and dinner times for the vegans and non-vegans were 7:23 and 7:02 a.m. (*p* = 0.348), 13:47 and 14:40 p.m. (*p* = 0.095), and 19:22 and 18:43 p.m. (*p* = 0.200), respectively. The vegans ate all main meals (per week) significantly more often than the non-vegans (breakfast: 6.9 ± 0.6 vs. 6.1 ± 1.8, *p* = 0.014; lunch: 7.0 ± 0.1 vs. 6.8 ± 0.7, *p* = 0.035; and dinner: 6.9 ± 0.3 vs. 6.7 ± 0.6, *p* = 0.046).

### 3.4. Cardiovascular Health Status and Linear Regression Analysis

The vegans had significantly lower levels of all lipids and BP than non-vegans, with the exception of HDL cholesterol. Both groups had plasma lipid profiles (i.e., LDL cholesterol levels ≤3 mmol/L and triglyceride levels ≤1.7 mmol/L) and BP (i.e., ≤129/84 mmHg) within recommended targeted values for the primary prevention of CV disease [47,48]. Cardiovascular health status is presented in Table 4.

Our results using multivariate regression analysis explained 47% of the variability in LDL cholesterol values. The regression model showed an inverse relationship between total fiber intake and LDL cholesterol (β = 0.012, *p* < 0.001). Furthermore, the partial correlation coefficients (β) indicated a decrease in LDL cholesterol by 0.12 mmol/L with each additional 10 g of total fiber intake. However, SFA intake was found to be positively associated with LDL cholesterol (β = 0.026, *p* < 0.001). Furthermore, the partial correlation coefficients (β) indicated an increase in LDL cholesterol by 0.26 mmol/L with each additional 10 g of SFA intake.

Furthermore, the model was adjusted for age, sex and smoking. However, only age additionally significantly explained the LDL cholesterol values. Finally, fiber and SFA intake alone explained 43% of the variance, while age added another 4% (β = 0.018, *p* < 0.001). The relationships between total fiber and SFA intake and LDL cholesterol is depicted in Figure 2.

### 3.5. Lifestyle Characteristics

#### 3.5.1. Transport Time, Everyday Sitting and PA

The vegans and non-vegans reported low transport time, relatively low weekly and weekend prolonged daily sitting, and high amounts of walking, moderate-intensity, vigorous-intensity and total PA. Importantly, there were no significant differences in any compared variables, except for moderate-intensity PA (Appendix A).

#### 3.5.2. Sleep Quality and Patterns

The vegans and non-vegans had good sleep quality and patterns (Appendix A). In addition, there was no significant difference between the groups in global sleep quality score. However, there was a significant difference in component 7 (i.e., daytime dysfunction: (i) trouble staying awake while driving, eating meals or engaging in social activity and (ii) having problems keeping up enough enthusiasm to get things done).

## 4. Discussion

### 4.1. Sociodemographics, Economic Status and Motive for Following the Dietary Pattern

In our study, we had more females than males, but we found no difference in the proportion of females between the groups. Furthermore, sociodemographics and economic status were also not significantly different between the vegan and non-vegan groups. Interestingly, both groups were mostly either married or living in an extramarital relationship. Most of them came from suburban and rural environments, and most of them finished primary/high school or had bachelor’s/university degrees and were employed/helping in family businesses or self-employed. Furthermore, there was a wide range of total family incomes, but the results suggested that most of those in the vegan and non-vegan groups belonged to the middle and upper classes, respectively. Finally, as expected for self-selected ‘health conscious’ groups, the smoking status showed that the majority never smoked (non-significantly higher in the vegan than non-vegan group; 78 vs. 55%, *p* = 0.071) or were former smokers. Only 4 and 14% of the vegans and non-vegans were current smokers.

According to a recent Slovenian national report, only 0.4% of adolescents and adults reported eating a vegan diet. However, all participants in this report were female [62]. Furthermore, the vegans and non-vegans in our study followed their dietary pattern primarily due to health motives, but significantly more vegans chose health as their primary motive. Interestingly, among the vegans, the second and third most important motives were satiety/no hunger, while among the non-vegans, these motives were reversed. Importantly, the ‘convenient’ motive was chosen significantly more often by the non-vegans than vegans.

An unhealthy diet within an unhealthy food environment poses tremendous challenges to eating in a healthy manner [63], regardless of whether this is a vegan or non-vegan dietary pattern. The decision to go on a healthy vegan or healthy non-vegan diet may have a number of underlying motives, and usually the strongest motives are primarily related to health reasons. Furthermore, researchers have found several common motives for choosing a vegan diet, from ethical and health benefits, BM management/appearance, eating to satiety, environmental concerns, and religious reasons [56,64,65,66]. On the other hand, the Finnish study suggested that eating motives are also associated with changes in diets, meaning that some eating motives might be being positively related to dietary change towards a vegan diet, whereas other motives could be interpreted as barriers to change (i.e., convenience and price motives) [63].

### 4.2. Anthropometric and Body Composition Status

Our results indicated that the vegans, compared to the non-vegans, had significantly lower levels of all measured variables (i.e., maximal lifetime BM and BMI, current BM, BMI, and BF%), with the exception of body height. Regardless, in terms of BMI, the non-vegans were considered to be in the overweight range (26.6 kg/m^2^) [36]. The possible reason for the body composition differences were not found in the comparable levels of PA, but likely possibilities were group differences in maximum BM/BMI, diet quality (i.e., total fiber and fat intake), and skipping meals. Importantly, the breakfast meal was the most often meal skipped by the non-vegans and may partially explain the measured differences in body composition status. In support of this hypothesis, a recent meta-analysis of 36 cross-sectional and nine cohort studies confirmed that skipping breakfast was associated with an increased risk of overweight/obesity [67].

As several researchers have noted, BMI is a poor index of the indication of total fat and has been criticized for its lack of sensitivity when distinguishing between fat mass and lean mass [68]. Due to the limitations of BMI, we combined it with BF% measurements in our study, which is especially important for smaller-scale observational studies and for people with potential sarcopenic obesity [69]. As noted, the vegans had significantly lower BF% than the non-vegans (19.3 vs. 25.8 kg/m^2^). However, obesity (i.e., related to BMI of 30 kg/m^2^) also corresponds to a BF% greater than 25% for men and 35% for women [37]. Since 67 and 55% of the vegan and non-vegan groups were female, we may conclude with great certainty that the non-vegan group was within the recommended range or at worst at the lower threshold of increased adiposity.

Furthermore, by comparing our results with the newest national data in a Slovenian report, while recognizing that we did not divide our samples by gender, the average assessed BF% (i.e., measured by bioimpedance of same manufacturer as in our study) of adult females was 33% (older females, 37.5%) and that of males was 24.9% (older males, 29%), compared with 19.3% and 25.8% in our vegans and non-vegans, respectively [62]. A study that compared the body composition (i.e., measured by bioimpedance) of 54 vegetarian Buddhist nuns with 31 omnivorous Catholic nuns found that BMI (22.6 vs. 20.7 kg/m^2^, *p* = 0.010) and BF% (24 vs. 21.8%, *p* = 0.037) were significantly lower among omnivores [70]. Notably, to understand the context of the obtained results, both groups of nuns in the study had very low total fat intake (15.6 vs. 13% of calories) and low animal fat (1.2 vs. 2.5% of calories) and cholesterol intake (10.8 vs. 70 mg/d) in the omnivore group. Moreover, the energy and protein intake among the studied groups were similar but included very low overall protein intake (11.7 vs. 11.5% of calories). Surprisingly, important for the interpretation of the obtained results, the intake of animal-based protein sources was unusually low for omnivores (0.6 vs. 2.4% of calories) [70]. Another study from Chinese researchers compared the body composition of 170 Buddhist monks on vegetarian diets (e.g., included only plant foods, with the exception of milk and eggs that were not popularly consumed) with 126 omnivore men recruited from a government administrative office in the same city. Vegetarians had a significantly lower BMI than omnivores (23.6 vs. 24.4 kg/m^2^, *p* = 0.042), while more subjects in the vegetarian group were within the normal BM range (61.8 vs. 56.4%) [71]. Last, the majority of observational studies have shown that the BMI of vegans was lower than that of controls (non-vegans); however, the BMI of controls was within the healthy weight range [72].

### 4.3. Dietary Intake Status

Dietary intake of key nutrients that are often critically compared between vegans and non-vegans are protein, fiber, free sugar, SFA, EPA and DHA, vitamin B_12_ and D, calcium, iron, iodine, and zinc. In our study, only protein and free sugar intake in units/d were similar between the groups. Importantly, most likely due to exclusion/inclusion criteria (i.e., ‘health conscious’ subjects), both diets resulted in high fiber intake and low free sugar and SFA intake, which is not something we would normally find in other cross-sectional studies [13,14,16,73], and reached targeted recommendations [44,45]. Furthermore, EPA plus DHA intake among the vegans was significantly lower than that among the non-vegans, and the vegans did not reach adequate intake compared with the recommendations [46]. Importantly, the analyzed vegan diet also included a large intake of seeds. Flax, chia, and hemp seeds are known to contain high amounts of alpha-linoleic acid (ALA). There is no reliable study on the conversion rate of ALA to EPA and DHA for a well-designed vegan diet, especially on a low-fat vegan diet (≤20% calories). Nevertheless, the body may convert up to 21% of ALA into EPA and up to 9% into DHA, and this process is more effective in females than males [74]. Micronutrients of concern (i.e., vitamin B_12_ and D, calcium, iron, iodine and zinc) were all significantly higher among the vegans than among the non-vegans, with the exception of iodine, which was similar.

Of great importance, vitamin B_12_ was supplemented by most vegans. Of note, 71% of the vegans consumed vitamin B_12_ in isolated form; furthermore, 27% of vegans consumed some amount of vitamin B_12_ within multivitamin dietary supplements. In addition, the non-vegans consumed sufficient vitamin B_12_ with foods alone (5.3 µg/d) [44]. Overall, vitamin D, calcium and iodine were low in both groups when compared with the recommendations [44]. In addition, vitamin D intake was not adequate for either group, presumably since the study was performed during the summertime, when vitamin D from supplementation for our geographic latitude (46° N) is not advisable [75]. Regardless, the non-vegans consumed 4.6 µg/d vitamin D from foods alone. Sodium and iodine intake could be appropriately solved with the inclusion of at least 3 g of iodized sea salt (1 g of iodized salt is enriched with 25 µg of potassium iodide) [76]; furthermore, also of appropriate iodine supplementation might be suggested. Based on the 3-day weighted dietary record used in our previous study with long-term vegans, fortified table salt in addition to the iodine in the food and multivitamin supplements, and not the routine intake of sea vegetables, was a major source of iodine [77].

Importantly, we can compare our results, especially in terms of frequently problematized nutrients (e.g., macronutrients: protein, fiber, SFA; micronutrients: vitamin B_12_ and D, calcium, iron, iodine and zinc), with only a few relevant studies with comparable dietary quality, separately for vegans and non-vegans. First, we compared dietary intake of our vegans (of note: without supplementation included) with dietary intake of Swiss vegans (n = 43) and with a study using a theoretically created 30-day whole-food, plant-based diet meal plan (‘WFPB diet’ study), both of which also assessed intake without including supplementation [15,78]. Protein intake in the Swiss and ‘WFPB diet’ studies was 65 g/d (11%) and 81 g/d (16%), respectively, vs. 92 g/d (16%) in our study. The dietary intakes in these two studied vegan groups also showed high fiber intake (52 and 70 g/day vs. 72 g/d in our study) and low SFA intake (7 and 3% vs. 3% in our study). Furthermore, vitamin B_12_ was not relevant for the comparison because in a vegan diet, it depended solely on dietary supplementation; vitamin D intake was low in the Swiss study and in our study. Low vitamin D intake with measurement through supplementation was expected due to the summertime study period, when the production of vitamin D in Slovenia via sun exposure is sufficient [75]. Interestingly, over one-fifth of our vegans supplemented with vitamin D_3_ dietary supplements. Assessed calcium intake in Swiss vegans and in the ‘WFPB diet’ study were relatively low as in our study (817 and 959 mg/d vs. 901 mg/d), whereas high iron intake was seen in all three studies (23 and 26 mg/d vs. 28 mg/d in our study) [15,78]. Zinc intake was assessed only among Swiss vegans (11.5 vs. 16.5 µg/d in our study) [15], whereas average iodine intake in these two studies was not provided (76 µg/d in our study without accounting for iodized salt or supplementation).

For non-vegan (i.e., also non-vegetarians) comparisons, we took the results from the Adventist Health Study 2 (AHS-2) with the use of dietary supplements and a study using a theoretically created 21-day ‘MyPlate’ meal plan taken from USDA Dietary Guidelines without the use of dietary supplements [12,78] and compared them with our results from non-vegans without supplementation. The estimated protein intake in the AHS-2 and ‘MyPlate’ study were 74.7 g/d (14.9%) and 96 g/d (19%), respectively, vs. 98 g/d (20%) in our study. Furthermore, the fiber intake among non-vegans in these studies was 29.8 and 28 g/d vs. 31.6 g/d in our study. Furthermore, SFA intake in all three studies was within the targeted recommendation (8.6 and 3% vs. 8.8% in our study) [44]. Vitamin B_12_ intake for all compared groups (7.1 and 6 µg/d vs. 5.3 µg/d in our study) was also within the recommended values, whereas vitamin D (6.1 and 10 µg/d vs. 4.6 µg/d in our study) was well below the targeted reference values [44]. Furthermore, calcium intake in these studies for non-vegans was 1072 and 1434 mg/d compared with 758 mg/d in our study, which was sufficient only in the ‘MyPlate’ study (i.e., compared with our valid recommendation) [44], while iron intake was 20 and 15 mg/d compared with 28 mg/d in our study, which was sufficient in all three studies [44]. Zinc intake was assessed only in AHS-2 (11.9 vs. 10.9 mg/d in our study) and may not be completely sufficient since both groups had high intake of whole grains, legumes and plant-based protein sources, which consequently interferes with zinc absorption due to high phytate intake [27,44]. Last, iodine intake was measured only in our study (65 µg/d). Overall, this comparison might not be fully justified in terms of actual circumstances because we would need to consider the replaced foods that would potentially displace supplementation in our non-vegans.

Furthermore, we estimated that 98% of vegans regularly consumed supplementation (71% used vitamin B_12_ in isolated form) compared with 87% of non-vegans. New national data in Slovenian reports estimated that 22.5–37.7% of adolescents and adults regularly consume (at least once per month) vitamins, minerals, multivitamins or omega-3 fats in the form of dietary supplements [62]. Moreover, these results are still more similar to ours when we compare only dietary supplements, since we found that 27% of vegans and 43% of non-vegans consumed only dietary supplements (i.e., multivitamins, excluding protein shakes/MR or sport drinks). In terms of proportion of vegans and non-vegans that regularly supplemented their diet, our results are in line with Finnish cross-sectional study, where 91% of vegans and 78% of non-vegans commonly used dietary supplements [13] and less consistent with the Danish cross-sectional study on vegans (65.7% of vegans used dietary supplements) and with the Swiss study on vegans (43% of vegans consumed dietary supplements containing vitamin B_12_) [11,15]. In contrast, in a German (RBVD) cross-sectional study on vegans and omnivores, 92% of vegans supplemented with vitamin B_12_ (97.2 and 33.3% of vegans and omnivores at least one dietary supplement), and the researchers found the average values of serum vitamin B_12_ and 25-hydroxyvitamin were 458 pg/mL and 68.6 nmol/L [17].

### 4.4. Cardiovascular Health Status and Linear Regression Analysis

Our results suggest that a well-designed vegan diet is associated with more favorable blood lipids and BP profile than a well-designed non-vegan diet. Vegans had significantly lower lipids than non-vegans, with the exception of HDL cholesterol. However, both groups had plasma lipids and BP within recommended targeted values [47,48]. Importantly, HDL cholesterol levels of our vegans and non-vegans (1.4 vs. 1.5 mmol/L) were within the ‘optimal’ 1–2 mmol/L range, which is associated with the lowest mortality risk [79]. Notably, both compared groups had high levels of PA, including walking and moderate-intensity PA, which are considered aerobic PA. In line with our results, aerobic PA can significantly reduce LDL cholesterol and increase HDL cholesterol [80].

Our results are of great importance since the most comprehensive research in Slovenia regarding primary CV disease prevention, where researchers, as part of national programs have tested 500,000 adults (25% of all Slovenians), have found elevated cholesterol levels in 69.2% of examined adults [81]. Furthermore, according to data from the National Institute of Public Health of Slovenia, almost 50% of adults have elevated BP [82]. In a recent meta-analysis of 31 observational studies comparing the effects of vegan and omnivorous diets on cardio-metabolic factors, the mean LDL cholesterol levels in vegans and omnivores was 2.36 mmol/L and 2.85 mmol/L, respectively (i.e., 0.66 mmol/L more than our vegans and no difference compared with our non-vegans). The report also included 29 studies where triglycerides were reported. The mean triglycerides in vegans and omnivores in this meta-analysis were 1.1 mmol/L and 1.24 mmol/L (0.3 and 0.16 mmol/L higher than our vegans and non-vegans). Finally, blood pressure was reported in 19 studies. The mean BP of omnivores was 121.8/75 mmHg; however, in non-Asian studies, systolic and diastolic BP was 6 and 3 mmHg lower in vegans, whereas no difference was found in Asian studies (contributing 82% of total vegan cohorts with BP) [72]. In our study, both systolic and diastolic BP were 7 mmHg lower among vegans than among non-vegans.

Although a well-designed vegan diet may result in an expectedly more favorable CV health status, the well-designed non-vegan diet in our study enabled CV health status within recommended values. However, the average LDL cholesterol in the non-vegans was 2.8 mmol/L, which is higher than the currently recommended levels for long-term preservation of cardiovascular health [83,84]. In brief, it was suggested that LDL cholesterol ≥2.6 mmol/L may be associated with subclinical atherosclerosis, even in the absence of other risk factors [83,84], which might become a serious health concern or even fatal later in life. The possible reason for the differences found in CV health status is the existence of differences in the quality of the dietary patterns. The most pronounced differences found between our groups were found in higher fiber and lower total fat and (zero) cholesterol intake among the vegans than the non-vegans, which was a typical finding in the above-described studies. Thus, we suggest at least two possible mechanisms. First, it is well known that higher fiber intake is associated with decreased CV diseases and mortality [85,86]. However, there are still not enough data in primary prevention settings about further benefits if fiber intake is higher than normally recommended levels and incorporated into well-designed non-vegan diets, as observed in our study or in the ‘MyPlate’ study. In partial support of this mechanism, in recent previous cross-sectional study, researchers measured lower total and LDL cholesterol status in subjects who were on a long-term high-fiber vegan diet than in those on a medium- or short-term diet [87]. Second, in a systematic review and meta-analysis of 15 trials, researchers found that dietary cholesterol significantly increased both total and LDL cholesterol when the daily intake did not exceed 900 mg [88]. Of note, cholesterol intake in non-vegans in our study was slightly above the recommended level (385 mg/d). Regardless, there is still insufficient evidence regarding primary prevention against CV diseases, particularly in the context of our study with the non-vegans, where concomitant fiber intake was well above 30 g/d (i.e., there is almost automatically no room for a large intake of harmful refined and (ultra)processed foods), SFA intake was within recommendations, and mean LDL cholesterol levels were 2.8 mmol/L.

Our results with multivariate linear regression showed that total fiber and SFA intake explained 43% of the variability in LDL cholesterol, while age added another 4%. First, each 10 g of total fiber intake was associated with a 0.12 mmol/L decrease in LDL cholesterol. This is in line with the balance of evidence that suggests a dose–response relationship with high intakes of fiber and lower risk for several non-communicable diseases, including CV disease [85,86]. Our results regarding higher fiber intake, also observed among the non-vegans, with reference to CV health status may be of importance, since the majority of the U.S. and European populations do not consume sufficient amounts of fiber [89,90]. Second, contrary to the total fiber intake and LDL cholesterol relationship, each 10 g SFA intake was associated with a 0.26 mmol/L increase in LDL cholesterol. Although SFAs may increase LDL cholesterol in most individuals, some SFA-rich foods (i.e., whole-fat dairy, dark chocolate and unprocessed meat) may not be associated with an increased risk for CV diseases [91]. Researchers have suggested that the health effects of foods cannot be predicted by their content in any nutrient group without considering the overall macronutrient distribution [91]. The major benefits of eating a high-fiber diet might be associated with a concomitant change in dietary pattern, which is lower in total cholesterol and SFA intake and higher in unsaturated fatty acid, mineral, folate, and antioxidant vitamin intake [92,93], as estimated in our study. Last, age and gender differences have also been associated with higher LDL cholesterol values [94], which was in line with our results. First, our study participants in both groups were older adults (46 and 57 years, for vegan and non-vegan subjects), while the majority of subjects in both groups were female (67 and 55%). Second, despite these facts, our subjects in both groups were able to control their body composition and CV health status with healthy and active lifestyles.

### 4.5. Lifestyle Status

Our study showed that there were no significant differences in lifestyle status (i.e., including smoking status) between the vegans and non-vegans. Importantly, both groups performed above the level of PA at which the highest number of health benefits have been reported to occur (6325 and 8147 MET minutes/week vs. recommended 3000 MET minutes/week) [95]. Furthermore, our vegans and non-vegans also outperformed the vigorous-intensity PA minimum recommendation (2160 and 2400 vs. recommended 1500 MET minutes/week) [51].

It is often assumed that the health benefits of vegan diets compared to non-vegan diets are related to vegan dieters being more likely to engage in healthier lifestyle choices [22,96]; however, we could not associate the possible differences between vegans and non-vegans in body composition and CV health status with lifestyle choices. However, perhaps the unusually high overall PA status of both groups may have been partly attributable to the fact that Slovenians were in partial ‘lock-down’ at the time of the study, which resulted in more spare time to be spent engaging in PA.

Regardless, our results of average weekly and weekend sitting times (5.4 vs. 4.2 h/d and 3.6 vs. 4.9 h/d for vegans and non-vegans, respectively) are in line with results found for the general adult Slovenian population [97]. However, researchers with the Slovenian National Institute of Public Health study also found that simple office work, intellectual, research and management-type work increased the average sitting time to 8.2 h/d. In conclusion, the shorter sitting times of our sample may have been indicative of a different organization of their lifestyle compared with the general population.

Given the results of the duration of transport to work (19 vs. 22 min/d for vegans and non-vegans), it seems that our sample was very much embedded into the local environment in terms of location of the business.

## 5. Strengths and Limitations

This is the first study that compared the nutritional and CV health status of lifestyle-matched ‘health conscious’ vegan and non-vegan subjects in Slovenia. Notably, the strength of our study is that it was performed during the COVID-19 pandemic period (partial ‘lock-down’), where subjects’ lifestyle status was still maintained at an enviable level compared with recent national data. In the study, we included all anonymously included subjects who completed the survey without missing data. In the final analysis, we precisely evaluated the actual dietary intake of free-living vegans and non-vegans from foods and from supplementation (combined and separately). Importantly, we also provided unique insight into ‘free sugar’ intake instead of only ‘added sugar’ or ‘total sugar’. Furthermore, the study invited and captured subjects from across the country (e.g., national representative), so the results may reflect the actual situation.

Since the study was designed during the pandemic period and had the aim of estimating the differences between ‘health conscious’ subjects, several limitations should be acknowledged. The main limitation is the nature of the study design (e.g., cross-sectional data, self-selected, and self-reported, but healthy and very active individuals); thus, the possibility of reporting errors remains, and therefore, our results may not be completely generalizable to individuals following general vegan and various forms of non-vegan diets. Of note, this was our only option during the pandemic where we were limited in our movement (i.e., ‘locked down’ within the municipality) for half of the research period. Nevertheless, the self-selected and self-reported nature of the study increases the possibility of reporting bias. Additionally, the number of participants was limited; however, many exclusion/inclusion criteria narrowed the representative research sample to one with which we unexpectedly captured more vegans than non-vegans and more females than males. In line with this observation, our sample was not divided by gender. Of note, vegans are being advised more than ever to regularly check their blood profile, while females are known to be more represented in nutritional studies. Last, in the final analysis, we did not include table salt intake, which is commonly used in food preparation and typically contributes to the intake of sodium and iodine; however, in the dietary intake analysis, we did consider enriched foods (i.e., from soya and grains), although these foods are not yet strongly present in Slovenia.

## 6. Conclusions

In summary, our results from a cross-sectional study show significant differences in dietary intake, body composition and CV health status of ‘health conscious’ vegans and non-vegans. With regard to sociodemographic, lifestyle, and other characteristics, we found no significant difference between vegans and non-vegans. Additionally, both groups chose health as the most important motive in deciding on their dietary pattern. On the other hand, vegans had a more favorable body composition status than non-vegans.

The dietary intake of both groups was on average based on unrefined and unprocessed whole-food diets. Overall, the vegan diet provided a more nutrient-dense diet than the non-vegan diet, with higher fiber and lower total cholesterol and SFA intake. However, both diets were insufficient in EPA and DHA (only vegans), vitamin D, calcium, sodium (only vegans) and iodine. The majority of subjects used selected supplementation, but there was no significant difference between the groups in energy or units/d of macro- and micronutrients. Importantly, non-vegans skipped all main meals more often than vegans, and the largest difference was with breakfast.

Furthermore, vegans exhibited a potentially more favorable CV health status than non-vegans. However, the CV health status for both groups was within the recommendations. In addition, in multivariate linear regression analysis, fiber and SFA intake alone explained 43% of the variability in LDL cholesterol, while age added 4%, for a total of explaining 47% of the variance.

Thus, the obtained results provide novel evidence that a well-designed vegan diet potentially enables a more favorable body composition and CV health status than a well-designed non-vegan diet among lifestyle-matched adult subjects. Although our study does not have the potential to suggest a causal relationship between a ‘health conscious’ vegan lifestyle and beneficial effects on body composition and CV health compared to a ‘health conscious’ non-vegan lifestyle, it is reasonably evident that a healthy version of the vegan and non-vegan lifestyle showed differences. Regardless, future studies should include larger samples, and long-term prospective randomized controlled studies with ‘health conscious’ vegans and non-vegans, where both diets are based primarily on unrefined dietary food sources.

## Figures and Tables

**Figure 1 ijerph-18-05968-f001:**
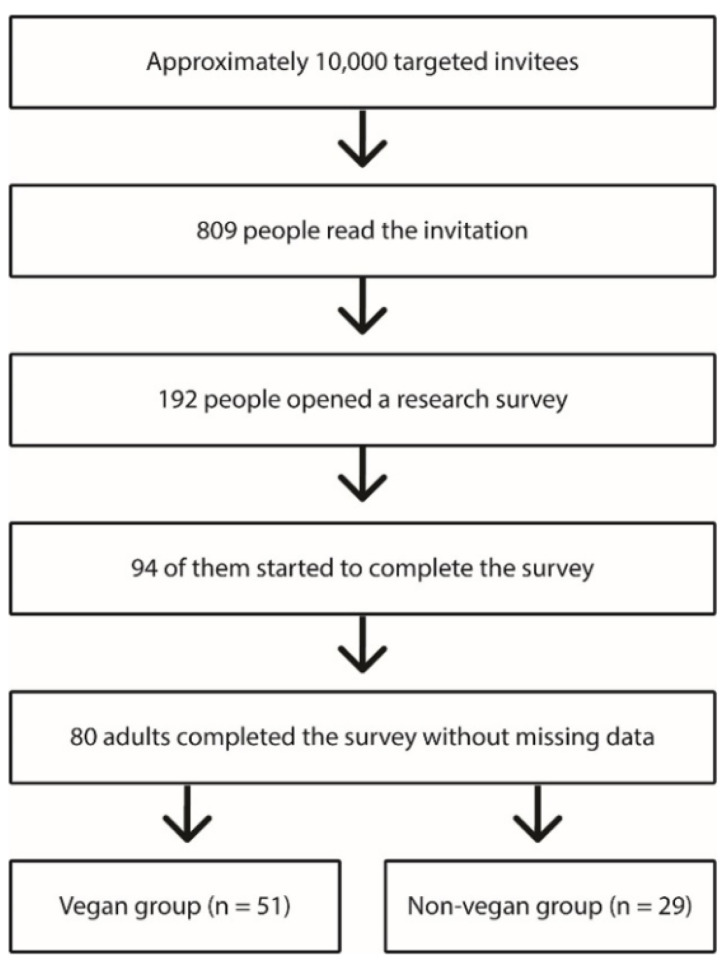
Recruitment process for participation in the study.

**Figure 2 ijerph-18-05968-f002:**
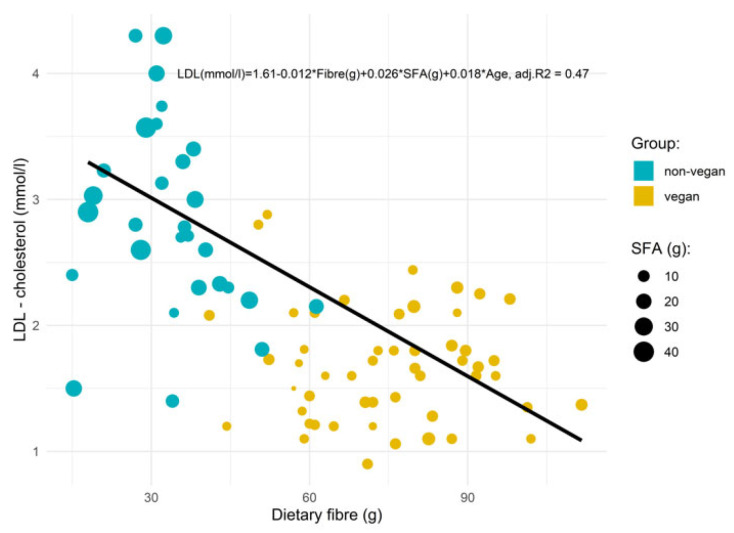
Multivariate regression relationship between fiber, SFA intake and age and LDL cholesterol.

**Table 1 ijerph-18-05968-t001:** Body composition status.

Parameter	Vegans (n = 51)	Non-vegans (n = 29)	*p*-values
Height (cm)	169.5 ± 9.1	168.7 ± 10.1	0.722
Maximal lifetime ^†^			
Body mass (kg)	76.8 ± 14.5	86.9 ± 19.6	**<0.001** *
BMI (kg/m^2^)	26.6 ± 3.7	30.3 ± 5.3	**0.001** *
Current			
Body mass (kg)	65.9 ± 11.6	76.1 ± 14.5	**0.002**
BMI (kg/m^2^)	22.8 ± 2.4	26.6 ± 3.6	**<0.001**
BF (%)	19.3 ± 7.3	25.8 ± 8.2	**<0.001** *

Data are means ± standard deviation (SD). Statistically significant values are written bold. ^†^ Maximal reported body mass (BM) that a participant reached at any time during their life. * A *t*-test was used; for other variables, the Mann–Whitney U-test was used.

**Table 2 ijerph-18-05968-t002:** Intake of energy and macronutrients.

Macronutrients (per Day)	Vegans	Non-Vegans	*p*-Value
Energy intake (kcal)	2399 ± 507	1998 ± 419	**<0.001**
Carbohydrates (g)	344 ± 85	212 ± 51	**<0.001**
(% E)	57 ± 5	43 ± 8	**<0.001**
Total sugars (g) ^TS^	74 ± 26	54 ± 21	**<0.001**
(% E)	12 ± 3	11 ± 4	0.334
Free sugars (g) ^FS^	9 ± 10	13 ± 14	0.329
(% E)	2 ± 2	3 ± 3	0.164
Starches (g)	116 ± 60	76 ± 41	**0.002**
(% E)	20 ± 9	15 ± 7	0.053
Dietary fiber (g)	75 ± 16	34 ± 10	**<0.001**
(% E)	6 ± 1	3 ± 1	**<0.001**
Fat (g)	54 ± 14	75 ± 25	**<0.001**
(% E)	20 ± 5	33 ± 7	**<0001**
SFA (g)	7 ± 2	20 ± 9	**<0.001**
(% E)	3 ± 1	9 ± 3	**<0.001**
MUFA (g)	13 ± 4	21 ± 8	**<0.001**
(% E)	5 ± 2	10 ± 3	**<0.001**
PUFA (g)	26 ± 8	18 ± 6	**<0.001**
(% E)	10 ± 3	8 ± 3	**0.021**
EPA + DHA (mg)	184 ± 288	527 ± 476	**<0.001**
Cholesterol (mg)	0	385 ± 507	**<0.001**
Protein (g)	97 ± 23	102 ± 32	0.652
(% E)	16 ± 2	20 ± 4	**<0.001**
(g/kg body mass)	1.5 ± 0.4	1.4 ± 0.4	0.178
Plant protein (g)	97 ± 23	43 ±15	**<0.001**
(% E)	16 ± 2	9 ± 2	**<0.001**
Animal protein (g)	0	60 ± 33	**<0.001**
(% E)	0	12 ± 6	**<0.001**
Alcohol (mg/day)	0	0.4 ± 0.9	**<0.001**
Water (L) ^w^	2.6 ± 0.6	2.2 ± 0.6	**0.004**

Data are means ± SD. Statistically significant values are written bold. % E = percentage of total energy intake (general Atwater energy conversion factors were used (kcal/g): carbohydrates and protein = 4, dietary fiber = 2, fat = 9, alcohol = 7) [60]. ^TS^ Total sugars: all monosaccharides and disaccharides: free sugars ^FS^ plus sugars naturally present in foods (e.g., lactose in milk, fructose in fruits) [61]. ^FS^ Free sugars: all monosaccharides and disaccharides added to foods and beverages by the manufacturer, cook or consumer (i.e., added sugars) plus sugars naturally present in honey, syrups, fruit juices and fruit juice concentrates (defined by the WHO [61] and adapted by the SACN [45]). SFA = saturated fatty acids; MUFA = monounsaturated fatty acids; PUFA = polyunsaturated fatty acids; LA = linoleic acid; ALA = alpha-linolenic acid; EPA = eicosapentaenoic acid; DHA = docosahexaenoic acid. ^w^ Water: from foods, beverages, and supplementation. For all variables, Mann–Whitney U-tests were used.

**Table 3 ijerph-18-05968-t003:** Intake of selected vitamins, minerals, and trace elements.

Micronutrients (per Day)	Vegans	Non-Vegans	*p*-Value
Vitamins			
Thiamine (mg)	2.8 ± 0.8	2.0 ± 0.7	**<0.001**
Riboflavin (mg)	2.0 ± 1.0	2.3 ± 1.4	0.440
Niacin (mg)	18 ± 9	24 ± 11	**0.006**
Pantothenic acid (mg)	7.4 ± 2.6	8.2 ± 3.6	0.319
Vitamin B_6_ (mg)	3.3 ± 0.9	2.1 ± 0.7	**<0.001**
Biotin (µg)	64 ± 26	86 ± 45	**0.042**
Folate (µg)	966 ± 341	468 ± 196	**<0.001**
Vitamin B_12_ (µg)	23 ± 72	5.6 ± 4.2	**<0.001**
Vitamin A (µg)	1.4 ± 1.0	0.9 ± 0.5	**0.036**
Vitamin C (mg)	273 ± 113	104 ± 74	**<0.001**
Vitamin D (µg)	7.0 ± 17.6	6.8 ± 5.4	**<0.001**
Vitamin E (mg)	16 ± 7	16 ± 9	0.799
Vitamin K (µg)	570 ± 374	195 ± 141	**<0.001**
Minerals			
Calcium (mg)	979 ± 281	826 ± 324	**0.038**
Magnesium (mg)	865 ± 190	541 ± 155	**<0.001**
Phosphorus (mg)	2090 ± 540	1898 ± 471	0.177
Potassium (mg)	4785 ± 1476	3682 ± 997	**<0.001**
Sodium (mg) ^†^	1069 ± 676	1756 ± 1277	**0.004**
Chloride (mg)	1731 ± 1012	2512 ± 1287	**0.004**
Trace elements			
Iron (mg)	32 ± 9	21 ± 6	**<0.001**
Iodine (µg) ^†^	112 ± 60	96 ± 58	0.238
Zinc (mg)	19 ± 5	13 ± 4	**<0.001**
Selenium (µg)	69 ± 24	104 ± 55	**0.006**

Data are means ± SD. Statistically significant values are written bold. ^†^ Sodium and iodine intake are from food and supplements only (i.e., without iodized salt). For all variables, Mann–Whitney U-tests were used.

**Table 4 ijerph-18-05968-t004:** Cardiovascular health status.

Parameter	Vegans	Non-Vegans	*p*-Value
Laboratory variables			
S-cholesterol (mmol/L)	3.3 ± 0.6	4.7 ± 0.9	<0.001 *
LDL cholesterol (mmol/L)	1.7 ± 0.4	2.8 ± 0.7	<0.001 *
HDL cholesterol (mmol/L)	1.4 ± 0.4	1.5 ±0.5	0.300
Triglycerides (mmol/L)	0.8 ± 0.3	1.4 ± 0.9	<0.001
Blood pressure (mmHg)			
Systolic	113 ± 11	120 ± 12	0.012 *
Diastolic	69 ± 8	76 ± 10	<0.001 *

Data are means ± SD. Statistically significant values are written in bold. * A *t*-test was used; for other variables, the Mann–Whitney U-tests were used.

## Data Availability

The data used to support the findings of this study are included within the article.

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
