# Peer review of "Nutritional, Cardiovascular Health and Lifestyle Status of ‘Health Conscious’ Adult Vegans and Non-Vegans from Slovenia: A Cross-Sectional Self-Reported Survey"

_ijerph, 2021, doi:10.3390/ijerph18115968_

Round 1

Reviewer 1 Report

Nutritional, cardiovascular health and lifestyle status of ‘health conscious’ adult vegans and non-vegans from Slovenia: A cross-sectional self-reported survey

The authors present a corss-sectional study addressing a very relevant topic. The study design is robust and the study outcomes are diverse and  the data seems to be reliably collected. Results are well presented and very detailed. There are some questions and additional explanations that the reviewer would like to know about. After the authors provide the information about these questions, also in the manuscript, the paper might be ready for acceptance.

Introduction:

Lines 46-48: is that statement supported by any reference? This is a very relevant point that cannot be just mentioned, but a bit more discussed (e.g. refs 16 and 17 could help?)

Line 67-70: why within the same scope of “healthy diet” or equivantly healthy diets between the vegan and the non-vegan one is recommended to have 20% of energy from fat and the other one is 30%? The same question for fibre. The reviewer believes that if authors are trying to evaluate the sole effect of the diet quality (this is, with matched groups in terms of lifestyle and.), both vegan and non-vegan diets should consist of the same nutrient quantities, so that the food-origin could be assessed as the main factor affecting the evaluated outcomes (BP, etc). Please, could you explain why you decided this? Which references did you use for such a choice? The reviewer is concerned about this decision

Methods:

This section is well structured, detailed and well explained.

Line 115-116: what is the difference between dietary choices and actual food intake?

Line 122: why did you target 10.000 people? As per power calculation for statistical analyses? Or based on a calculation that reaching 10.000 peopole would revert in approx.. 80 subjects recruitment? Then, why is the final sample 80 subjects?

Inclusion and exclusion criteria: very well defined and selected adequate parameters. However the definition of healthy diet in line 134 should be more detailed: “as much as possible” should be further explained in terms of ranges: e.g. unrefined sources less than once a day or a week. Also a definition of whole-food or unrefined sources should be provided and possibly including examples. This is a critical aspect that would help to understand the significance of the results.

Statistical analysis: this section is well explained and the reported analyses are robust and adequate for the type of research questions. The reviewer wonders why “free sugars” intake was not used in any of the statistical models to assess its impact on some study outcomes, as this is a very relevant nutrient.

Results: results are very well presented, in a very detailed manner. The reviewer is satisfied to see that in this type of study focused on the diet, reliable and wide range of information is presented. However, the reviewer suggests that, if possible, information about frequency of consumption is provided in terms of food groups: dairy, fruit, vegetables, nuts, etc…

Figure 2: it is very self-explanatory and represents the main result. But it is blurry and cannot read well. Please replace.

Discussion: it is very long. The reviewer suggests to summarise all the sub-sections within.

Line 730-731: very relevant indeed. But in methods the reviewer could not see the explanation on how this estimation of free sugars or added sugars was made. Suggest to incorporate this relevant information.

Reviewer 2 Report

Thank you for the opportunity to review this manuscript. A lot of attention and resources are spent on plant-based diet. The paper is very fluent, clear and easy to read. Related work is critically analyzed. This study focus is very important. Generally, the results and discussion were clearly articulated, however I have 2 minors, but important changes"

1. Figure 1 is unreadable. Please prepare new figure.

2. Please add a short paragraph on the definition BMI.

Reviewer 3 Report

This study compared vegan and non-vegan health status and diet among health-conscious adults. It is well designed and seems to be useful to the reader. Please refer to the following a few comments.

Would you like to explain detail about L113-115 “Each participant self-identified their current diet by choosing from one of several options (i.e., omnivore, vegetarian (“lacto”, “ovo”, “pesco”, “lacto + ovo”) or vegan).” Because, even those who have little understanding of vegetarians and vegans can easily understand the meaning of each meal.

Figure 1 & 2 is unclear. It's hard to read. Please prepare a clear figure.

Regarding nutrients intake, nutritional research often adjusts energy (e.g.  /1000 kcal intake). What is the reason for using the absolute amount in this study? Is it necessary to compare nutrients intake that energy is adjusted?
